# Amino Acids in Cancer and Cachexia: An Integrated View

**DOI:** 10.3390/cancers14225691

**Published:** 2022-11-19

**Authors:** Maurizio Ragni, Claudia Fornelli, Enzo Nisoli, Fabio Penna

**Affiliations:** 1Center for Study and Research on Obesity, Department of Biomedical Technology and Translational Medicine, University of Milan, 20129 Milan, Italy; 2Department of Clinical and Biological Sciences, University of Torino, 10125 Turin, Italy

**Keywords:** amino acid, cancer metabolism, cachexia, nutrition, supplement

## Abstract

**Simple Summary:**

Cancer metabolism is an emerging field of investigation aimed at identifying cancer cell vulnerabilities in order to define novel anti-cancer therapeutic approaches based on interventions that modulate the availability of specific nutrients. Amino acids (AAs) are used by cancer cells as both building blocks for protein synthesis required for rapid tumor growth and as sources of energy. The current review aims to describe the most relevant alterations of AA metabolism that could be targeted by either AA deprivation or AA supplementation to limit tumor growth. In parallel, the reader will understand how AA availability mainly relies on and impacts cancer-host metabolism, eventually leading to a wasting paraneoplastic syndrome called cachexia. The above-mentioned AA-based interventions are here discussed also in view of their impact on the tumor host, in an attempt to provide a broader view that can improve our understanding of the patient outcome.

**Abstract:**

Rapid tumor growth requires elevated biosynthetic activity, supported by metabolic rewiring occurring both intrinsically in cancer cells and extrinsically in the cancer host. The Warburg effect is one such example, burning glucose to produce a continuous flux of biomass substrates in cancer cells at the cost of energy wasting metabolic cycles in the host to maintain stable glycemia. Amino acid (AA) metabolism is profoundly altered in cancer cells, which use AAs for energy production and for supporting cell proliferation. The peculiarities in cancer AA metabolism allow the identification of specific vulnerabilities as targets of anti-cancer treatments. In the current review, specific approaches targeting AAs in terms of either deprivation or supplementation are discussed. Although based on opposed strategies, both show, in vitro and in vivo, positive effects. Any AA-targeted intervention will inevitably impact the cancer host, who frequently already has cachexia. Cancer cachexia is a wasting syndrome, also due to malnutrition, that compromises the effectiveness of anti-cancer drugs and eventually causes the patient’s death. AA deprivation may exacerbate malnutrition and cachexia, while AA supplementation may improve the nutritional status, counteract cachexia, and predispose the patient to a more effective anti-cancer treatment. Here is provided an attempt to describe the AA-based therapeutic approaches that integrate currently distant points of view on cancer-centered and host-centered research, providing a glimpse of several potential investigations that approach cachexia as a unique cancer disease.

## 1. Introduction

Cancer metabolism is a broad and emerging field of study at the frontier of cancer discovery, aimed at understanding tumorigenesis, tumor progression, and disorders of cell metabolism, as well as for designing new prospective therapies [1]. The dysregulation of metabolic circuitries occurs at several levels beyond cancer cells, affecting tumor stroma, the immune system, and eventually the whole host, resulting in uncontrolled tumor growth, immunoescape, and cachexia, respectively. Understanding the cellular tissue and systemic metabolic alterations will likely uncover, on the one side, exploitable metabolic vulnerabilities for reducing tumor growth and improving anti-tumor therapies, and on the other side, specific nutrient host deficiencies which, upon supplementation, will counteract cachexia and support a more effective fight against the cancer. The objective of the current review is to focus on amino acid (AA) metabolism in both cancer cells and the tumor host in order to highlight potential AA-based therapeutic approaches that integrate the currently distant points of view on cancer-centered and host-centered research.

### 1.1. Impact of AAs on Cancer Metabolism and Mitochondrial Function

The elevated biosynthetic activity of cancer cells is supported by several strategies aimed at increasing growth. Oncogenic mutations allow tumor cells to sustain angiogenesis, evade apoptosis, enable constitutive proliferation signaling by tyrosine kinase receptors, and to escape from growth suppression signaling [2]. Moreover, profound metabolic alterations also confer to cancers the ability to optimize substrate utilization; one typical example of metabolic reprogramming is the Warburg effect (aerobic glycolysis), the observation that cancer cells convert glucose to lactate even in an oxygen-rich environment, dates back to almost 100 years ago [3]. The Warburg effect is not exclusive to cancer cells, but also occurs in normal, non-cancerous, rapidly dividing cells, such as activated macrophages and lymphocytes, haematopoietic stem and progenitor cells, or during angiogenesis, and this confirms the growth advantage conferred by this metabolic adaptation [4]. Initially ascribed to defective mitochondria, this apparently inefficient metabolic reprogramming is now hypothesized to have the aim, rather than ATP production, of maximizing carbon delivery to the cell’s anabolic pathway for the synthesis of biomass. In both living organisms and in culture, cells in fact rarely experience shortages of glucose, which is always kept at high/constant levels in the culture medium or bloodstream; this means that proliferating cells do not have a real need to maximize ATP synthesis from glucose [5]. Therefore, by engaging glucose into aerobic glycolysis, cancer cells avoid its complete catabolism to ATP in mitochondria; this spares glucose carbon that can be used for generating acetyl-Coa, glycolytic intermediates, and ribose for the synthesis of fatty acids, non-essential amino acids (NEAA), and nucleotides, respectively [6]. Since the major part of these pathways require functional mitochondria, this implies that, in proliferating cancer cells, mitochondria are not dysfunctional; however, as a consequence of aerobic glycolysis, they are rather used as a biosynthetic organelle for the synthesis of glucose-derived lipid and NEAA, instead of ATP. Furthermore, cancer cells are also able to efficiently utilize mitochondria for directing other cell substrates for macromolecule synthesis and, in this process, amino acids (AA) play a pivotal role [7]. The role of AAs as constituents of proteins and/or signaling molecules involved in the regulation of macroautophagy, the process of endosomal/lysosomal recycling of cellular components, or as activators of biosynthetic cell pathways through mammalian target of rapamycin (mTOR), has been the subject of many other comprehensive reviews [8,9,10]. However, AAs show an intimate connection with mitochondrial function. One such example is glutamine (Gln), which, together with glucose, is the main molecule utilized by the majority of mammalian cells in culture.

### 1.2. Cancer Mitochondria Support Biosynthesis of Macromolecules through Glutamine-Dependent Anaplerosis

Since the early discovery that HeLa cells consume Gln from 10 to 100 orders of magnitude more than other amino acids [11], and that Gln is used as a major source of cell energy, rather than for incorporation into proteins [12], the observation that transformed cells display a high rate of Gln consumption has been confirmed in several other cancer cell types, such as glioblastoma, ovarian, pancreatic, and breast cancer [13,14,15,16]. Through mitochondrial glutaminolysis, Gln is utilized to provide both carbon and nitrogen for anabolic reactions; in particular, through the process of anaplerosis, which leads to the replenishment of the pools of metabolic intermediates of the tricarboxylic acid cycle (TCA) in times of high energy requirements [17], Gln is a major carbon source for the synthesis of proteins, nucleotides, and lipids. In this pathway, Gln is first converted into glutamate (Glu) via mitochondrial glutaminase (GLS), and then glutamate dehydrogenase (GDH) convert glutamate into α-ketoglutarate (α-KG); alternatively, a second pathway of conversion of Glu in α-KG involves the activity of both mitochondrial and cytosolic transaminases such as aspartate transaminase (AST), alanine transaminase (ALT), or phosphoserine transaminase (PSAT). The routing of glutamate to the dehydrogenation or to the transamination pathway depend on the metabolic status of the cell and has important metabolic consequences; although the prevalence of transamination over dehydrogenation has been positively correlated with proliferation rate [18] and glucose levels, both pathways enable the optimization of amino acid uptake and consumption with growth and biomass production, with transamination being prevalent in times of glucose abundance, while the glutamate dehydrogenation pathway has been shown to be induced during glucose shortages, thus enabling cell survival [19,20]. Furthermore, sequential glutaminase and glutamate dehydrogenase reactions produce ammonia (NH_3_) and ammonium (NH_4_^+^), respectively, which is considered a cell toxic by-product which requires scavenging through urea; however, ammonia is also a major diffusible autophagy inducer [21], which could therefore help cancer cells increase fitness by eliminating damaged and potentially toxic macromolecules and organelles. Moreover, in breast cancer, extracellular ammonia synthesized in the liver can be recycled by tumor cells through reductive amination by GDH to form glutamate which is then used for macromolecule biosynthesis [22], and Gln-derived ammonia can also promote sterol regulatory element binding protein (SREBP) mediated lipogenesis and support tumor growth [23]. A graphical representation of the above-described metabolic scenario is provided in Figure 1. Glutamate transamination, on the contrary, does not produce ammonia, but generates the NEAAs aspartate (Asp), alanine (Ala), and serine (Ser) which, together with α-KG, serve as anaplerotic substrates in the TCA cycle, thus promoting biosynthesis. In this process, Oxaloacetate (Oaa) produced by α-KG from glutaminolysis condenses with pyruvate-derived Acetyl-CoA to produce citrate at the citrate synthase step; citrate is then not committed to the isocitrate dehydrogenase step (which is instead involved in reductive carboxylation—see below), but is instead exported to cytosol where ATP-citrate lyase reforms Acetyl-CoA, which is then used for malonyl-Coa synthesis and de novo lipogenesis. Glutaminolysis flux also generates reduced nicotinamide adenine dinucleotide (NADH) and reduced nicotinamide adenine dinucleotide phosphate (NADPH), as reducing equivalents for the electron transport chain (ETC) and both redox balance and lipogenesis, respectively. Beyond glutaminolysis, which is considered as the oxidative branch of Gln metabolism, the alternate metabolic fate of Gln is through the reductive carboxylation pathway, which involves the engaging of α-KG in a “reverse” mitochondrial TCA cycle through NADPH-dependent isocitrate dehydrogenases 1 and 2 (IDH1 and IDH2) and therefore producing citrate for lipid biosynthesis [24,25,26]. This pathway seems to be predominant in cancer subjected to hypoxia, displaying constitutive hypoxia-inducible factor 1 alpha (HIF1α) activation, or have defective mitochondria [27,28] and has been confirmed to occur also in vivo [28,29].

### 1.3. Mitochondrial Electron Transport Chain and AA Metabolism

Beyond the evidence coming from glutamine/glutamate metabolism, several other studies clearly demonstrate that the function of mitochondria in sustaining cell proliferation goes beyond the simple ATP synthesis, and that biomass production and macromolecule synthesis depend on to the strict connection and crosslink between mitochondrial activity and AA metabolism. A striking example on how mitochondrial function and AA metabolism are reciprocally intertwined comes from two studies which show that the most essential function of the ETC in proliferating cells is to provide electron acceptors to support Asp biosynthesis which, in turn, is required for the synthesis of purine and pyrimidines [30,31]. By recycling NAD^+^, ETC provides oxidized cofactor for malate dehydrogenase 2 (MDH2) which, in turn, produces Oaa that is used by mitochondrial aspartate transaminase (GOT2) for the synthesis of Asp. However, when ETC is dysfunctional, the ratio NAD^+^/NADH decreases, and Asp synthesis is switched to the reductive carboxylation of Gln to citrate which, through ATP-citrate lyase, produces Oaa to drive Asp synthesis. This suggests that, in proliferating cells, the high NAD^+^/NADH ratio maintained by mitochondrial respiration is essentially used for Asp/nucleotide synthesis, rather than for ensuring constant TCA cycle/OXPHOS function. Treatment of acute myeloid leukemia (AML) cells with IACS-010759, a complex I inhibitor, reduced cell viability, decreased NAD^+^/NADH ratio, and, among AAs, exclusively reduced Asp levels while increasing Gln utilization, thus confirming that the interconnection between mitochondrial activity, redox state, and Asp synthesis is extended to cancer cells [32]. This relationship has also been recently confirmed in vivo; treatment of mouse neuroblastoma xenografts with IACS-010759 affected ETC activity, redox state and Asp metabolism, as well as activation of reductive carboxylation of Gln. In addition, treatment of xenografted mice with metformin, a commercial available antidiabetic drug and complex I inhibitor, dose-dependently inhibited tumor growth, together with a decrease in intratumor NAD^+^/NADH ratio and Asp levels [33,34]. The evidence of reprogramming of the cancer mitochondrial function to support Asp synthesis for survival comes not only from studies based on pharmacological inhibition of complex I, but also from genetic disruption of ETC components, which resemble mitochondrial alterations in tumors; inactivating mutations of mitochondrial succinate dehydrogenase (SDH) are frequent in cancers and increase susceptibility to the development of different tumors and, as such, SDH has tumor suppressor functions. Immortalized kidney mouse cells deficient in SDH become addicted to extracellular pyruvate for proliferation, which, through carboxylation to Oaa, is diverted to Asp synthesis. Notably, SDH ablation in absence of pyruvate also decreased the NAD^+^/NADH ratio, thus confirming that, in cancer cells, mitochondrial function is strictly connected to Asp biosynthesis through the modulation of its redox state [35]. Moreover, a very recent paper revealed a close relationship between ETC, redox state, and Gln utilization for biomass production: in 143B cells, respiration inhibited by both mitochondrial DNA (mtDNA) depletion of parental ρ0 cells and chemical ETC inhibition, the reduced mitochondrial NAD+/NADH ratio reversed mitochondrial GOT2 as well as succinate and malate dehydrogenases to promote mitochondrial oxidation of NADH to NAD+; this, in turn, enabled GDH-dependent Gln anaplerosis to support cell proliferation [36]. Beyond Asp, mitochondrial NAD^+^ regeneration has been shown to be essential also for the synthesis of Ser, which is another carbon source for the production of nucleotides; in fact, withdrawal of Ser from culture medium increased sensitivity of melanoma cell lines to the anti-proliferative effects of both rotenone or metformin; furthermore, Ser deprivation depleted cells of purine nucleotides and rotenone treatment further increased their depletion [37]. Alteration of Ser synthesis also occurs in response to cells depleted of mtDNA; this causes bioenergetic failure and induction of activating transcription factor 4 (ATF4), a master transcription factor which regulates cell response to AA starvation during the integrated stress response (ISR) [38]. mtDNA depleted cells display increased Ser synthesis and decreased Ser consumption, thus reflecting elevated reliance on serine for survival [39]. Ser, together with glycine (Gly), is also involved in one-carbon (1C) metabolism, a process shared by both mitochondria and cytosol which, through the folate cycle, leads to the production of 1C methyl units for several biochemical pathways such as purine and pyrimidine biosynthesis, and synthesis of methionine (Met), as well as providing the moiety for methylation reactions for epigenetic control of gene expression [40]. Being a central hub for a plethora of anabolic pathways, cancer cells can therefore become reliant on generation of 1C units from both Gly and Ser. Ser can be converted to Gly by the cytosolic or mitochondrial serine hydroxymethyltransferase SHMT1 and SHMT2, respectively, with the transfer of Ser-derived 1C unit to tetrahydrofolate (THF), generating methylene-THF which, in turn, is required for purine and pyrimidine biosynthesis. As a result, cancer cells have been shown to both avidly consume extracellular Ser and also to strongly depend on endogenous Ser synthesis, displaying an elevated Ser flux [41,42,43,44]. Ser can be synthesized from the glycolytic intermediate 3-phosphoglycerate (3-PG) through the enzyme phosphoglycerate dehydrogenase (PHGDH), which is in a genomic locus of copy number gain in both breast cancer and melanoma, and whose protein levels are increased in 70% of estrogen receptor (ER)-negative breast cancers [42,44]. The overexpression of PHGDH in tumors allows elevated rates of de novo Ser biosynthesis which, as a result, increases the biosynthesis and metabolic pathways associated with the folate pool, amino acid, lipids, and redox regulation [44]. However, in tumors, this metabolic adaptation requires a diminished flux of pyruvate toward mitochondrial oxidation, thus redirecting upstream glycolytic intermediates, such as 3-PG, to Ser and Gly synthesis; in many cancer cells, this is accomplished by preferential expression of the M2 isoform of pyruvate kinase (PKM2), which displays lower kinase activity compared to PKM1, thus promoting the accumulation 3-PG and other glycolytic intermediates which are precursors for biosynthesis of Ser, as well as nucleotides, amino acids, and lipids required for proliferation [45]. Notably, Ser is an allosteric activator of PKM2, and this allows cancer cells to rapidly increase Ser biosynthesis in response to its environmental shortage since, when Ser levels fall, the decrease in PKM2 activity allows more glucose-derived carbon to be diverted into serine biosynthesis [46]. Although Ser can be synthesized by both SHMT1 and SHMT2, the mitochondrial isoform is strongly upregulated in cancers and has been shown to have a stronger impact on cancer metabolism [42,47]. Mitochondrial 5,10-methylenetetrahydrofolate production by SHMT2 provides methyl groups for mitochondrial tRNA modification, which are required for proper mitochondrial translation and function [48,49]. Furthermore, SHMT2 is required for complex I assembly, and mice deficient in SHMT2 display embryonic lethality and defective mitochondrial respiration [50,51].

### 1.4. Targeting AA Metabolism as Anti-Cancer Strategy

The data herein reported indicate that the connection between AA metabolism and cancer mitochondrial function goes beyond the well-known role of AAs as mitochondrial substrates and can also explain why tumors are dependent on some AAs. Notably, this addiction also includes AAs usually classified as non-essential such as Gln, Ser, and Asp, therefore implying that the mitochondrial reprogramming of cancer metabolism also re-defines and crosses the distinction between essential and non-essential AAs. This AA dependency highlights a metabolic vulnerability, which could be exploited for a highly specific anticancer therapy aimed to starve or deplete cancer cells of selected amino acids, or to block crucial AA metabolic pathways [52].

*GLN* Glutaminase inhibition has been long proposed as a main cancer therapeutic target, and several Glnase inhibitors have been developed, of which the most studied are 6-diazo-5-oxo-L-norleucine (DON), bis-2-(5-phenyl acetamido-1,2,4-thiadiazol-2-yl) ethyl sulfide (BPTES), and CB-839; however, since glutaminolysis is not an exclusive feature of cancer cells (see above), low specificity/high toxicity concerns and poor solubility issues have limited their employment in clinic, and only CB-839 (Telaglenastat) has since been entered in a clinical trial and is currently in phase II [53].

*ASN* Asparagine (Asn) depletion by means of the bacterial enzyme asparaginase (ASNase) is presently the only approved anticancer treatment based on an AA-depleting approach and has been long used as an anticancer strategy, especially toward pediatric acute lymphoblastic leukemia (ALL) [54,55]. Contrary to most cells, leukemia cells express low levels of asparagine synthetase (ASNS), which render them highly dependent on Asn, thus making ASNase treatment effective [56]. Although several toxic side effects have also been also reported, the majority of them are manageable, but ASNase is almost exclusively used only in ALL [57].

*ARG* Inhibition of cancer cell proliferation by means of arginase (ARGase) treatment was reported almost 70 years ago [58]; since then, the depletion of Arginine (Arg) through ARGase or arginine deiminase (ADI) administration has been constantly explored as an anticancer therapy, also supported by the finding of absent or low expression of argininosuccinate synthetase (ASS1) in several tumors, especially those associated with chemoresistance and poor clinical outcome, such as hepatocellular carcinoma (HCC), melanoma, mesothelioma, pancreatic cancer, prostate cancer, renal cell carcinoma, sarcoma, and small cell lung cancer [59,60,61,62], which result in Arg dependence. While Arg-deprived, non-cancer cells undergo quiescence and cycle arrest at G0/G1 phase, cancer cells starved for Arg continue instead to DNA synthesis, leading to unbalanced growth and ultimately to cell death [63]. Currently, ARGase, as a PEGylated derivative, is employed in clinical trials for the treatment of HCC, melanoma, prostate adenocarcinoma, and pediatric acute myeloid leukemia (AML), while trials with PEGylated ADI are ongoing for the treatment of small cell lung cancer, melanoma, AML, HCC, and mesothelioma [64].

*SER, GLY* The importance of targeting Ser, Gly, and Met metabolic pathways, which are strictly interconnected to folate 1C metabolism, is highlighted by the long-known use of antifolate drugs as chemoterapics [65]; methotrexate (MTX), the most used antifolate, deplete cells of tetrahydrofolate and is routinely used for the treatment of multiple cancers. However, MTX treatment has some concerns of toxicity [66]. Since, in tumors, the majority of 1C units derive from Ser, many efforts to develop inhibitors of Ser biosynthesis to inhibit cancer 1C metabolism, alone or in combination with other antifolates, have been put forward [67]. Pyrazolopyran, an herbicide compound, which was originally shown to inhibit plant SHMT, has been shown to inhibit human SHMT1 and to induce cell death in lung cancer cells [68]. An optimized pyrazolopyran derivative, by targeting both SHMT1 and SHMT2, blocked proliferation of colon, pancreatic, and several B-Cell derived malignancies, which displayed different grades of sensitivity according to their proficiency in folate metabolism and glycine uptake, thus confirming amino acid vulnerability as a key target for anti-cancer drug development. Furthermore, incubation of many cancer cells in medium lacking Ser/Gly greatly increased the sensitivity of the cells to the PHGDH inhibitor PH755, while PH755 treatment in vivo to xenografted mice significantly potentiated the anti-cancer response of dietary serine/glycine restriction, thus underscoring the effectiveness of a combined dietary and chemotherapeutic approach [69].

## 2. Amino Acid Supplementation as Anticancer Therapy: Targeting Multiple and Complex Metabolic Networks

The majority of anticancer strategies aimed at targeting the AA metabolism and/or the AA dependency of tumor cells are based on AA starvation; less is known about AA supplementation which has also shown to be beneficial [70]. Dietary starvation of a single AA is not a simple task and strategies to limit Gln, Asp, or Ser/Gly intake had a limited positive response due to compensatory metabolic mechanisms [71,72]; clinically speaking, nutritional enrichment of selected AAs is certainly a more accessible objective. Supplementation of branched-chain amino acids (BCAA) is long-known to have favorable effects in patients with hepatocellular carcinoma [73,74,75,76] and is recommended to cirrhotic patients according to the guidelines of the American Association for the Study of Liver Diseases (AASLD) and the European Association for the Study of the Liver (EASL) [70]. Despite its well-known role in fueling cancer proliferation, Gln supplementation, by both increasing its dietary content and supplementation in drinking water, blocked melanoma growth in mice by affecting the epigenetic marks of oncogenic gene expression [77]. Furthermore, dietary supplementation of histidine (His) by enhancing its catabolic flux and depleting tumor cells of tetrahydrofolate, which is also a major target of MTX, increased the sensitivity of leukemia xenografts to MTX chemotherapy [78]. Moreover, a more recent report showed that an AA-defined diet enriched in EAAs decreased tumor growth in mice [79]. Mechanistically, dietary enrichment of EAAs activated BCAA catabolism, inhibited glycolysis and mTOR signaling and induced cancer cell apoptosis through a ATF4-mediated ER stress response, which derived from an intracellular Glu shortage. These results suggest that supplementation of AAs can lead to inhibition of cancer cell proliferation through multiple mechanisms.

The finding that supplementation of EAA can lead to a decrease in levels of other AAs [79] highlights the complexity of intracellular AA metabolic pathways, which are known to be closely linked to each other, showing reciprocal cross-talk, sharing many metabolic intermediates, and also responding with different cues in a strong context-dependent manner [52]. A deep analysis of human metabolic networks showed that AAs are strongly interconnected within specific groups: Gly, Ser, Ala, and Thr; cysteine (Cys) and Met; valine (Val), leucine, and isoleucine (Ile) display the most interconnected pathways [80]. Furthermore, AAs are connected to 39 different cellular processes; notably, these include not only well-characterized AA pathways such as protein synthesis, biomass production and membrane transport, but also specific metabolic pathways like fatty acid oxidation, metabolism of glutathione, nicotinamide adenine dinucleotide, and sphingolipids [80]. Within these, AAs share about 1139 metabolites which are involved in AA metabolism, both directly and indirectly, through interactions with other downstream pathways. Among the top metabolites linked to AA metabolism, NADPH, ATP, NADP^+^, ADP, and NAD^+^ play a pivotal role in regulating other core cellular metabolic pathways: this indicates a ubiquitous role for AA in the control of cell metabolism [80]. According to their multifaceted role in cell physiology, many key regulators of cancer AA signaling show a double-face nature; the two isoforms of Glnase, GLS1 and GLS2, are usually over- and under-expressed in cancers, with GLS2 considered as a tumor suppressor, in opposition to the established role of GLS1 as a pro-oncogenic gene [53,81]. Consistently, the PI3K/Akt pathway and mTORC2, two main regulators of cell growth and proliferation, have been shown to be inhibited by exogenously supplemented BCAA [82]. It is noteworthy that, through systems biology analysis, these pleiotropic functions of AAs and their different connections to cell metabolites have been recently shown to have the potential to be exploited for the treatment of multifactorial diseases [80,83]. The intricate and non-univocal impact of either starving or supplementing AAs in cancer cells is summarized in Figure 2.

Cancer is a complex disease. Beyond being influenced by genetic, physiological, and environmental factors, it also has, especially in advanced stages, several outcomes for the whole organism, which are the result of a network of metabolic alterations which involve different organs’ dysfunctions. As such, an effective therapeutic strategy aimed at targeting the multiple complications of cancer will require a comprehensive approach. By engaging and targeting different and complex metabolic pathways, AA-based approaches are therefore supposed to impinge more effectively on the multiple dysfunctions developing in the different tissues affected by cancer complications. Cachexia, which lead to weakness, muscle atrophy, systemic inflammation and weight loss can be considered as a paradigm of the complex multifactorial condition associated with cancer [3] and is therefore particularly suitable for nutritional and dietary AA-based approaches.

### 2.1. Cancer-Induced Cachexia

Cancer cachexia is a complex multiorgan syndrome that affects more than half of all cancer patients (50–80%) and is the direct cause for about 20% of cancer deaths [84,85]. It is characterized by involuntary loss of body weight and adipose tissue that cannot be fully restored by conventional nutritional support, as well by systemic inflammation, anorexia, asthenia, fatigue, metabolic alterations, and muscle wasting, resulting in progressive functional impairment [86,87]. These changes induce detrimental effects on the immune system, increasing susceptibility to infections or other complications, affecting the efficacy and the tolerance to chemotherapy and surgery and, finally, impacting on a patient’s quality of life [87,88]. Furthermore, 30 to 90% of cancer patients show malnutrition, characterized by reduced appetite, calorie intake, and changes in taste, accompanied by malabsorption, maldigestion and dysmetabolism, and diminished food intake. This condition may be exacerbated by antineoplastic treatment-related side effects which act on the intestinal epithelium of the host and are responsible for nausea and vomiting (see previous chapter) [89].

Sarcopenia is a gradual and progressive decline in skeletal muscle mass and functional capacity, considered as an age-associated syndrome but also occurring earlier in life in association with chronic diseases, including cancer [90]. Loss of skeletal muscle mass and strength are considered the main clinical criteria for the diagnosis of sarcopenia. The occurrence of the syndrome is confirmed by different techniques including dual-energy X-ray absorptiometry (DXA), computerized tomography (CT), and magnetic resonance imaging (MRI), together with the evaluation of walking speed and handgrip strength, but the absence of standardized evaluation methods in clinical practice could make it difficult to discriminate sarcopenic from non-sarcopenic individuals [86,91]. Moreover, sarcopenia is often associated with other underlying clinical conditions that affect muscle wasting including frailty, cachexia, and sarcopenic obesity, increasing the risk of adverse health-related outcomes such as physical disability, postoperative infections, hospitalization, and institutionalization, resulting in poor quality of life and mortality [86,91,92,93].

Cancer cachexia is associated with metabolic alterations. In particular, patients show a simultaneous hyper-metabolism, hyper-catabolism, and hypo-anabolism, resulting in a decrease in biomolecule synthesis and a concomitant increased degradation, aggravating the weight loss and causing impaired energy metabolism [85]. For these reasons, cachexia could be considered as an ‘energy wasting’ disorder in which the homeostatic control of energy intake and expenditure is lost and depends on the type of tumor and its stage [84,85]. Impaired protein turnover is a general hallmark of the energy waste typical of cancer cachexia, both in patients and in experimental models. In fact, there is an activated muscle proteolysis, accompanied by hypo-anabolism and mitochondrial alterations (impaired biogenesis, dynamics, and degradation) [94]. In cancer cachexia, AAs captured by the tumor and other metabolically active tissues such as the liver are obtained from lean tissues, in particular the skeletal muscle, thanks to the activation of two main proteolytic systems, namely the proteasome and the autophagic-lysosomal pathway induced by inflammation-dependent transcription factors, such as FoxO1/3 and NF-κB [95]. Previous studies demonstrated in the muscle of tumor-bearing animals and in cancer patients a hyper-activation of the autophagic system, inflammation and increased reactive oxygen species (ROS) [96,97]. The pro-oxidant species damage the mitochondria, leading to further ROS production and stimulating mitophagy, affecting mitochondrial abundance in the muscle [96,97,98,99]. Since mitochondria represent the main producers of energy required for contraction, alterations of their homeostasis impair muscle function [97]. In parallel, it was reported that antioxidant enzymes, such as superoxide dismutase (SOD), catalase, and glutathione peroxidase (GPx), are up-regulated in the muscle of tumor-bearing animals in the attempt to counteract the oxidative insult, albeit not enough to maintain the redox balance, while others reported a down-regulation of the same enzymes, further promoting oxidative stress [99].

So far, although several drugs are being tested in clinical trials; none of them proved effective enough to be used in common practice. Most of the compounds tested in cancer patients so far are based on orexigenic and anabolic compounds [100,101], the former aimed at reversing anorexia and increasing the food intake, the latter at stimulating the signaling pathways controlling muscle protein synthesis and at repressing the signals activating muscle protein catabolism. The current research aims to reverse muscle wasting, targeting several aspects of the syndrome, i.e., reducing energy expenditure, fatigue, anorexia, and systemic inflammation, improving the patient’s quality of life and physical state. Beyond this, one of the goals is to understand how the molecular alterations underlying muscle wasting impinge on metabolic and oxidative capacity and mitochondrial function, impairing muscle metabolism and not only muscle mass [102].

One fundamental prerequisite for making any anti-cachexia therapeutic intervention effective is to avoid the occurrence of malnutrition. The European Society for Clinical Nutrition and Metabolism (ESPEN) releases continuously updated guidelines on the management of cancer patients from the nutritional standpoint [103]. The best way to increase protein and energy intake is by food, but unfortunately this is not always possible and or sufficient [104]; appetite stimulants or dietary supplement hardly restore an optimal nutritional status and only partially reduce fat and skeletal muscle loss. AAs, as building blocks of proteins, represent for mammals the only source of nitrogen and the main component of skeletal muscle mass and, as described above, are also used as TCA cycle intermediates for producing ATP [105]. AAs are involved in the regulation of key physiological pathways in the body and their homeostasis is altered in various pathological conditions characterized by mitochondrial dysfunction and oxidative stress [105]. According to the ESPEN, a protein intake lower than 1 g protein/kg/day, not reaching a daily energy request of 25 kcal/Kg, and a nitrogen optimal uptake from protein between 1.2 and 2 g/kg/day is associated with decreased physical function and an increased risk of mortality [89,100,106]. Circulating AA levels are altered in cancer patients due to tumor demand of AAs, especially essential amino acids (EAAs), Gln, Gly, and Asp [89]. In particular, variations in plasma AAs were found in cancer patients depending on the occurrence of metabolic alterations, metastasis, anorexia, malnutrition, and weight loss; for these reasons, plasma AAs were considered as biomarkers for diagnosis and screening of cancer and a potential therapeutic target for improving protein synthesis and consequently muscle wasting [89,105]. More specifically, high levels of circulating AAs were described in breast cancer, while low levels were found in patients with gastric and colorectal cancers, as well as decreased levels of Arg in pancreatic and lung cancers [89].

### 2.2. Nutritional (AA-Based) Anti-Cachexia Supportive Interventions

In line with the commonly described AA shortage, many studies exploited the possibility of supplementing single AAs or mixtures of essential AAs as a therapeutic strategy in conditions characterized by oxidative stress, catabolic state, or altered energy balance, such as sarcopenia, sepsis, cardiac and metabolic diseases. Various compounds were studied, such as β-hydroxy-β-methylbutyrate (HMB), Gly, Leu, Arg, Gln, or AA mixtures [89,107,108]. A detailed description of the potential for each compound against cancer cachexia follows and is summarized in Figure 3. The ample evidence of AA efficacy in preclinical animal models of cachexia, however, does not align with a corresponding abundance of clinical trials in cancer patients, where most of the AAs tested were provided along with anti-inflammatory and antioxidant mixtures, not allowing for the discrimination of the AA contribution.

### 2.3. HMB

HMB is an endogenous product of Leu metabolism normally present in foods such as avocado, grapefruit, and catfish. It derives from Leu oxidation into ketoisocaproate, 95% of which is metabolized into Coenzyme A and the remaining 5% is converted into HMB [106]. Many supplementations that involve HMB alone or in combination with other AAs (Arg and Gln) were tested, and found to be safe and well tolerated in cancer patients [107,108]. In particular, HMB has been proposed to be effective in reducing inflammation and muscle proteolysis, and in stimulating protein synthesis through mechanistic target of rapamycin (mTOR) complex 1 (mTORC1), with beneficial effects in the maintenance of muscle mass and function in the elderly [86,89,91,105,108]. Many parameters were reported to be improved, including skeletal muscle mass, postoperative hospitalization, and complications insurgence. Moreover, these studies observed positive results regarding anticancer treatment-related side effects, tumor response, and hospitalization [91,107].

### 2.4. GLY

Gly, the simplest non-essential AA, is required for protein synthesis as well as being a precursor of fundamental molecules such as purines, heme, creatine, NADPH, and glutathione. In particular, it is involved in the regulation of redox homeostasis, modulating the balance between the two forms of glutathione, the oxidized (GSSG) and the reduced one (GSH) [109]. More specifically, Gly synthesis is stimulated as a mechanism of defense against mitochondrial dysfunction and oxidative stress; indeed, its supplementation is able to enhance NADPH and glutathione content, reducing ROS production and the expression of genes involved in inflammation and muscle macrophage infiltration [109]. Many studies demonstrated that intracellular Gly levels decrease during aging and result lower in association with frailty and in experimental mouse models of muscle wasting [109]. In tumor-bearing mice, Gly administration was able to reduce tumor growth, preventing muscle wasting and preserving muscle strength, suggesting a potential therapeutic use in cancer cachexia [109].

### 2.5. LEU

Leu is an essential amino acid that belongs to branched-chain amino acids, well known in literature to strongly stimulate skeletal muscle protein synthesis, modulating directly the activity of mTORC1 [110]. Positive effects of Leu supplementation were observed in animals fed with a high fat diet, with attenuation of mitochondrial dysfunction, fat mass, and hyperglycemia [105]. A few studies performed in animal models of cancer cachexia have shown that a diet rich in Leu was able to stimulate, in a dose-dependent manner, protein synthesis in skeletal muscle, resulting in reduced loss of lean body mass and muscle wasting [111]. These data were supported by improvements in muscle strength, physical performance, and inflammatory status [111]. Simultaneously, skeletal muscle mitochondrial biogenesis was ameliorated after Leu supplementation in tumor-bearing rats, together with the increased expression of markers related to oxidative phosphorylation and energy production [111]. Unfortunately, recent evidence suggested an increase of tumor growth rate with a Leu-rich diet, promoting cancer development and for these reasons limiting the clinical potential [111].

### 2.6. ARG

Arg is a semi-essential AA important for immune function and cell regeneration. 25–30% of the total daily Arg derives from food and the remaining 70–75% can be synthetized endogenously starting from citrulline or from protein turnover [89,106]. During cachexia or other catabolic and inflammatory conditions, Arg levels are decreased even without alterations in the biosynthesis [106]. In cancer patients, Arg supplementation demonstrated immunomodulating properties together with improvement in survival and malnutrition [89,106].

### 2.7. GLN

Gln is a semi-essential AA necessary for cellular function and immune cell modulation [89,106]. Remembering that Gln is also abundantly used by cancer cells for supporting their metabolism and growth (see above), its use as a supplement may have a dual impact on cancer (stimulating the growth) and on the host (counteracting cachexia). Experimental data seem to discard the first hypothesis since, using the experimental model of Walker-256 tumor-bearing rats, a promising role for Gln in improving energy balance and preventing tumor growth and cancer-induced cachexia was suggested [112,113]. In parallel, clinical trials in cancer patients highlighted that oral Gln supplementation is able to restore glutathione levels with beneficial effects on the antioxidant status and the immune system [89,106].

### 2.8. Essential AAs

Recent data showed that supplementation with an enriched mixture of branched-chain AAs (BCAAem), a subset of EAAs that comprises Leu, Ile, and Val, integral components of skeletal muscle proteins and powerful stimulators of protein synthesis, increases nitric oxide (NO) production via endothelial NO synthase (eNOS), inducing the expression of PGC-1α, a transcription coactivator and main regulator of mitochondrial biogenesis (reduced in cancer cachexia) and of genes involved in the ROS defense system including superoxide dismutase 1 (SOD1), superoxide dismutase 2 (SOD2), catalase and glutathione peroxidase (GPx1), decreasing oxidative damage and promoting survival [89,105,114]. As a result of PGC-1α increased expression, both glycogen accumulation and fatty acid oxidation improve, preserving mitochondrial metabolism and muscle fiber size, and ameliorating physical endurance and motor coordination in middle-aged mice, suggesting a potential role in the treatment of sarcopenia and cancer cachexia [105,114]. Recent evidence demonstrated that BCAAem was able to preserve skeletal muscle, mitigating muscular dystrophy in an experimental model of Duchenne Muscular Dystrophy (mdx mice), and clinical trials confirmed an improvement of physical performance, muscle mass, and strength in sarcopenic patients [105]. Another relevant study proposed supplementation with BCAA, Tyr, and Cys in order to improve lean muscle mass in cachectic patients with advanced cancer [89,91,101]. Despite some limitations, for example the numerosity, the lack of a control group, and the heterogeneity of patients according to cancer type, ROS reduction and improvements in strength and quality of life were observed [89,91].

Given that AA supplementation could be insufficient to obtain clinically relevant improvements on muscle mass and function, two novel AA formulations containing EAA-BCAA cofactors (citric acid, malic acid, and succinic acid) and precursors of Krebs cycle were explored in experimental models of age-related disorders and in general conditions characterized by mitochondrial dysfunction and altered catabolic state, resulting as more effective than the BCAAem alone [105]. The two mixtures differ in the percentage of each component. The first one (called a5) was able to promote the ROS defense system and mitochondrial biogenesis, preventing cardiac damage induced by doxorubicin; the second one (called PD-E07) was able to improve the expression of PGC-1α and the enzymatic activities of mitochondrial respiratory complexes in the skeletal muscle of mice characterized by age-related muscular alterations [105].

## 3. Concluding Remarks and Open Challenges

The selected evidence reported in this review highlights multiple aspects of AA biology in either the cancer itself or in the cancer host, meaning that any conclusion based on the study of only one of these two components is of limited value in its translation into a cure for the cancer disease in humans. Beyond this critical point, the narrative of AA relevance in cancer diagnosis and treatment provided here is definitively non-exhaustive. Several potential fields where deeper AA investigation may provide useful knowledge and tools against cancer, translating into innovative clinical applications, are described below.

*Biomarkers*. AA plasma levels may serve as disease biomarkers, guiding diagnosis and the choice of a potentially more effective treatment. As recently shown [115], the metabolomics era will soon produce breakthrough discoveries, unpredictable using the previous generation of targeted analyses. In this context, experts in cancer and host metabolism must interact to correctly interpret the significance of circulating AA levels resulting from both cancer and host metabolic activities.

*Supplementation* vs. *diet*. Although whole foods look more effective than oral nutritional supplements, at least in head and neck cancer patients undergoing cachexia [116], AA supplementation provides a precise dosage, allowing adjustments according to response (either by biochemical assessments or by disease follow-up). More comparative clinical trials are needed to define the best strategy.

*AA supplementation during chemotherapy*. Similarly to the dualism described in this review, AA may improve the host’s ability to tolerate chemotherapy and to support or even stimulate tumor immunity as a combined treatment with immunotherapy in the emerging field of immunonutrition [117].

*AA and microbiota in cancer patients*. Microbiota dysbiosis has been reported in both preclinical animal models and in cancer patients, resulting in altered AA metabolism [118,119]. AA supplements may impact on the microbiota, helping controlling tumor growth and cachexia.

In conclusion, personalized (P4) medicine is the only reasonable way of introducing AA-based supportive oncology strategies which, by treating the cancer patient as a whole (i.e., by fighting the cancer and supporting the host), will likely improve the outcome of several incurable cancers. However, fasting the tumor to impair cellular proliferation remains a goal to pursue. Consistently, Fasting Mimicking Diets (FMD) were proven effective in maximizing the chemotherapy response [120]. In this context, AA supplementation may accelerate the recovery after FMD-induced weight loss by shortening the interval between the therapy cycles, given that the best refeeding strategy after FMD is still unknown and deserves further research. Similarly, a recent report shows an attractive alternative strategy for starving cancer cells by means of cold exposure [121], suggesting that nutrient deprivation may be obtained with alternative approaches, avoiding feeding restrictions.

## Figures and Tables

**Figure 1 cancers-14-05691-f001:**
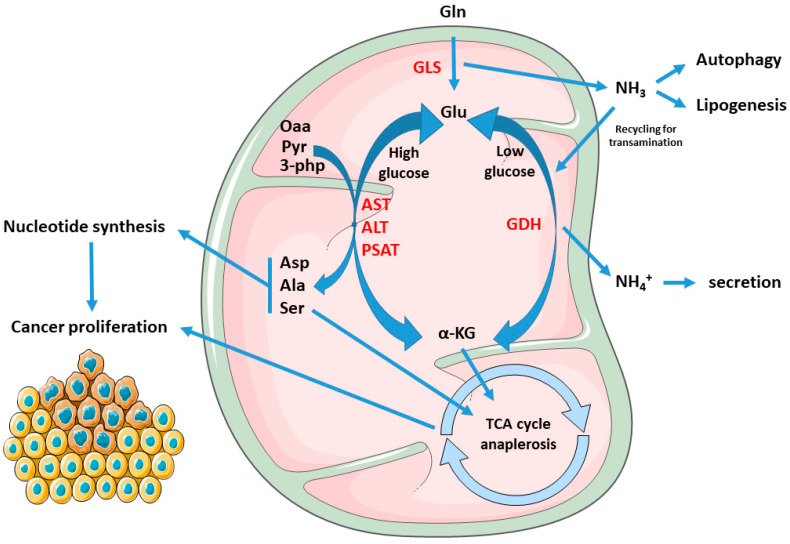
Rewiring of the metabolic fate of AAs in cancers: Gln- derived Glu yield α-kg through both transamination or dehydrogenation, by means of AST, ALT, PSAT, or GDH, respectively. The first pathway prevails in conditions of glucose abundance, resulting in production of the NEAAs Asp, Ala, and Ser, and enabling proliferation. Asp is used in nucleotide synthesis while α-kg, Ala, and Ser can feed the TCA cycle. On the other hand, GDH is activated during glucose scarcity or cell quiescence, supporting cell survival and stress resistance. Furthermore, GLS-derived ammonia in cancer can stimulate autophagy, lipogenesis, or being re-incorporated to Glu synthesis, which can be re-routed for transamination and biomass production. Abbreviations: Gln, glutamine; Glu, glutamate; Asp, aspartate; Ala, alanine; Ser, serine; GDH, glutamate dehydrogenase; GLS, glutaminase. AST, aspartate transaminase; ALT, alanine transaminase; PSAT, phosphoserine transaminase; Oaa, oxaloacetate; Pyr, pyruvate; 3-Php, 3-phosphohydroxypyruvate; α-KG, alpha-ketoglutarate.

**Figure 2 cancers-14-05691-f002:**
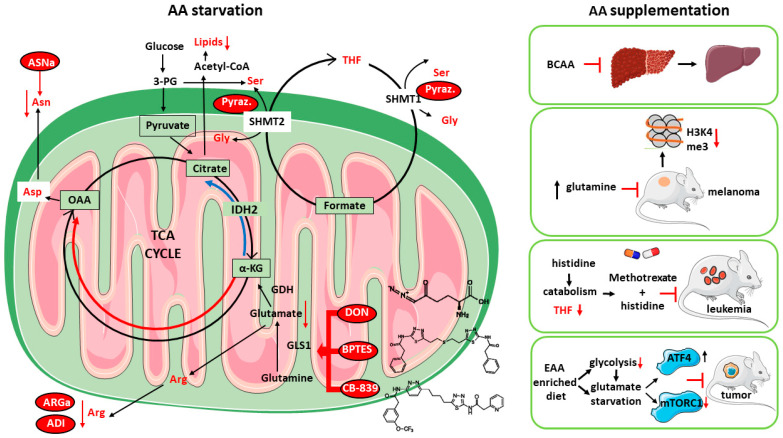
Both amino acid starvation (**left**) and supplementation (**right**) are valuable anti-cancer strategies. (**Left**) panel: mitochondrial glutamine consumption in cancer feeds TCA cycle in both oxidative (red) and reductive (blue) directions which, by providing carbon source for the synthesis of precursors, increases cancer biomass. Interfering with glutaminase (GLS1) activity with the inhibitors DON, BPTES, or CB-839 leads to the depletion of key anabolic intermediates such as lipids and aminoacids. Anti-cancer enzymatic approaches through asparaginase (ASNase), arginase (ARGase), or arginine deaminase (ADI) also lead to the decrease in blood levels of indispensable tumor AAs. Furthermore, interfering with serine or glycine metabolism with the dual SHMT1/SHMT2 inhibitor pyralozopyran also results in depletion of tetrahydrofolate (THF), a key metabolite essential for cancer growth. (**Right**) panel: BCAA supplementation is beneficial in cirrhotic patients, while diet with high gln content blocks melanoma xenografts in mice by reprogramming the tumor epigenetic control of gene expression. Histidine supplementation, by increasing His catabolism and depleting THF, increases methotrexate sensitivity in leukemia-bearing mice, while an EAA-enriched diet impairs tumor growth in mice by inhibiting glycolysis and mTORC1 signaling by means of an EAA-induced ATF4 activation through shortage of glutamate. Abbreviations: 3-PG, 3-phosphoglycerate; IDH2, isocitrate dehydrogenase 2; SHMT1/SHMT2, serine hydroxymethyltransferase ½; GDH, glutamate dehydrogenase; α-KG, alpha-ketoglutarate; ATF4, activating transcription factor 4; mTORC1, mammalian target of rapamycin complex1; H3K4me3, lysine-methylated histone 3; pyraz, pyrazolopyran. The icons used were obtained at: https://smart.servier.com/ (accessed on 10 November 2022).

**Figure 3 cancers-14-05691-f003:**
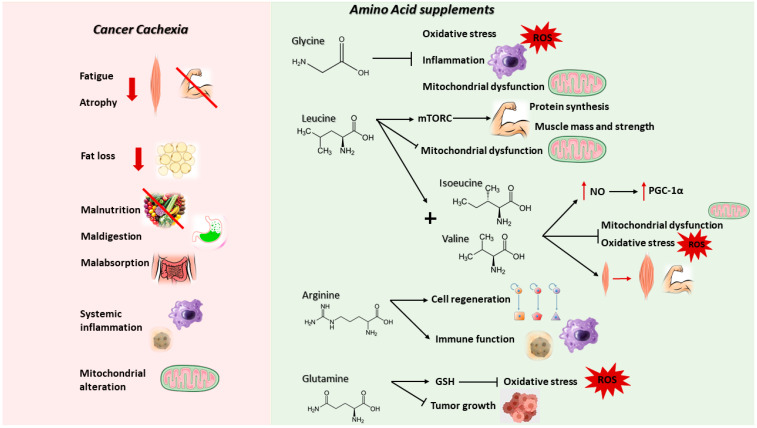
Main alterations underlying cancer cachexia (**left**) and the impact of specific AA supplementation in counteracting such manifestation or promoting beneficial anti-tumor effects (**right**). A detailed description of the anti-cachexia action is provided in the following text. The icons used were obtained at: https://smart.servier.com/ (accessed on 10 November 2022).

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
