# Peer review of "Amino Acids in Cancer and Cachexia: An Integrated View"

_cancers, 2022, doi:10.3390/cancers14225691_

Round 1

Reviewer 1 Report

The manuscript was prepared very well. The introduction section justifies the purpose of the study. I congratulate the authors for the preparation of the manuscript

However, I have the following comments:

·       There are several sentences and paragraphs without reference, add them.

·       What has led you to select only the amino acids that are described in the study? Why haven't they selected others? clarify this point.

·       Given that L-glutamine is an amino acid that helps to recover muscle mass, hasn't I considered a section on restoring muscle function in patients with cachexia?

·       These patients with cancer + cachexia usually have impaired metabolism, that's why you talk about leucine, wouldn't it be more convenient to use its metabolite HMB?

·       I recommend including, albeit briefly, a section on methodology, with the searches, terms and method of selecting the included studies. In addition, the objective of this narrative review could be clearly indicated.

·       Would an application or recommendation of the studied amino acids be possible?

·       What is new in this manuscript?

·       Include a section on strengths and weaknesses

·       In the Conclusion section, state the most important outcome of your work. Do not simply summarize the points already made in the body — instead, interpret your findings at a higher level of abstraction. Show whether, or to what extent, you have succeeded in addressing the need stated in the Introduction (or objectives).

Reviewer 2 Report

In the present manuscript, the authors provide a comprehensive summary on how aminoacid metabolism is rewired in cancer cells and discuss potential anticancer treatments by targeting aminoacid metabolism. A special focus is given to dietary interventions affecting both tumor and host aminoacid and protein metabolism in the context of tumor progression and tumor-induced cachexia. Overall, the manuscript is well written and structured and the authors have done an excellent job to cover most of the key findings in the field (with few exceptions listed below).  

Minor points:

1. The manuscript will benefit from adding an additional figure to summarize aminoacid metabolism rewiring in cancer cells.

2. The authors must check and fix their references section, there are many references that are not correct. For instance: reference 22 ( is not Am. J. Trop. Med. Hyg 2006, it is Nature 2011), reference 26 (is not Elsevier 2014, it is Cell Metabolism 2013), reference 29 (is not Physiol. Behav. 2017, it is Cell 2015)...

3. The section of "Mitochondrial Electron Transport Chain and AA metabolism" would be improved by including a recent study by Altea-Manzano et al. (PMID:36327975) describing the reversal of mitochondrial MDH2 to enable glutamine anaplerosis. 

4. In the section "Targeting AA metabolism as anticancer strategy", the authors could mention the role of inhibiting PHGDH in the context of Ser/Gly starvation (Tajan et al., PMID: 33446657).

5. When discussing AA supplementation as anticancer therapy, the authors make several statements with no clear source for the information provided, as no references are listed (see for instance lines 299-308). Furthermore, reference 36 included in line 308 does not seem to match the content discussed in the previous lines. 

6. Lastly, this manuscript would be further improved by adding a section (or few paragraphs) discussing the molecular mechanisms regulating tumor-induced cachexia (or host tissues wasting) in the context of aminoacid metabolism (induction of systemic autophagy, increased proteasomal activity, etc), besides explaining nutritional anti-cachexia interventions. 
